# HIV proviral genetic diversity, compartmentalization and inferred dynamics in lung and blood during long-term suppressive antiretroviral therapy

Aniqa Shahid[1,2], Bradley R. Jones[2,3], Julia S. W. Yang[4], Winnie Dong[2], Tawimas Shaipanich[5], Kathryn Donohoe[5], Chanson J. Brumme[2,6], Jeffrey B. Joy[2,3,6‡], Janice M. Leung[4,5‡], Zabrina L. Brumme[1,2‡]*

1 Faculty of Health Sciences, Simon Fraser University, Burnaby, Canada, 2 British Columbia Centre for Excellence in HIV/AIDS, Vancouver, Canada, 3 Bioinformatics Program, University of British Columbia, Vancouver, Canada, 4 Centre for Heart Lung Innovation, University of British Columbia, Vancouver, Canada, 5 Division of Respiratory Medicine, Department of Medicine, University of British Columbia, Vancouver, Canada, 6 Department of Medicine, University of British Columbia, Vancouver, Canada

‡ These authors are co-senior authors on this work.
* zbrumme@sfu.ca

**Data Availability Statement:** The nucleotide sequences reported in this paper are available in GenBank (accession numbers: proviral DNA:

## Abstract

The lung is an understudied site of HIV persistence. We isolated 898 subgenomic proviral sequences (*nef*) by single-genome approaches from blood and lung from nine individuals on long-term suppressive antiretroviral therapy (ART), and characterized genetic diversity and compartmentalization using formal tests. Consistent with clonal expansion as a driver of HIV persistence, identical sequences comprised between 8% to 86% of within-host data-sets, though their location (blood vs. lung) followed no consistent pattern. The majority (77%) of participants harboured at least one sequence shared across blood and lung, sup-porting the migration of clonally-expanded cells between sites. The extent of blood proviral diversity on ART was also a strong indicator of diversity in lung (Spearman's ρ = 0.98, p<0.0001). For three participants, insufficient lung sequences were recovered to reliably investigate genetic compartmentalization. Of the remainder, only two participants showed statistically significant support for compartmentalization when analysis was restricted to dis-tinct proviruses per site, and the extent of compartmentalization was modest in both cases. When all within-host sequences (including duplicates) were considered, the number of com-partmentalized datasets increased to four. Thus, while a subset of individuals harbour somewhat distinctive proviral populations in blood and lung, this can simply be due to unequal distributions of clonally-expanded sequences. For two participants, on-ART provi-ruses were also phylogenetically analyzed in context of plasma HIV RNA populations sam-pled up to 18 years prior, including pre-ART and during previous treatment interruptions. In both participants, on-ART proviruses represented the most ancestral sequences sampled within-host, confirming that HIV sequences can persist in the body for decades. This analy-sis also revealed evidence of re-seeding of the reservoir during treatment interruptions. Results highlight the genetic complexity of proviruses persisting in lung and blood during

OM963156 - OM964037, OP346862 - OP346877; HIV RNA: OM964038 - OM964339). The four custom R scripts mentioned in the methods can be found at: https://github.com/brj1/HIVCompartmentalization. A source file containing the data plotted in Fig 1A and 1B, Fig 2B and 2D, Fig 3B and 3D, Fig 4B and 4D, Fig 5B and 5C, Fig 6C and 6D, Fig 7C and 7D, S1B and S1D Fig, and S2B Fig is provided in S1 Data file.

**Funding:** This work was supported by the Canadian Institutes of Health Research (CIHR) through a project grant (PJT-159625 to Z.L.B. and J.B.J.) and a focused team grant (HB1-164063 to Z.L.B.). This work was supported by the Martin Delaney 'BELIEVE' Collaboratory (NIH grant 1UM1AI26617 to Z.L.B.), which is supported by the following NIH Co-Funding and Participating Institutes and Centers: NIAID, NCI, NICHD, NHLBI, NIDA, NIMH, NIA, FIC, and OAR. This work was also supported by the Martin Delaney 'REACH' Collaboratory (NIH grant 1-UM1AI164565-01 to Z.L.B.), which is supported by the following NIH co-funding Institutes: NIMH, NIDA, NINDS, NIDDK, NHLBI, and NIAID. A.S. and B.R.J. are supported by CIHR Doctoral Research Awards (stipend support). J.M.L is supported by a Health Professional Investigator Award from the Michael Smith Foundation for Health Research, and by a Tier 2 Canada Research Chair in Translational Airway Biology (salary support). Z.L.B. is supported by a Scholar Award from the Michael Smith Foundation for Health Research (salary support). The funders had no role in study design, data collection and analysis, decision to publish, or preparation of the manuscript.

**Competing interests:** The authors have declared that no competing interests exist.

ART, and the uniqueness of each individual's proviral composition. Personalized HIV remission and cure strategies may be needed to overcome these challenges.

## Author summary

HIV persists in the body despite long-term antiretroviral therapy (ART). Much of our knowledge about the HIV reservoir comes from studying blood proviruses on ART, but a fundamental question is whether these are distinct from those in tissue. The lung could theoretically engender genetically distinctive HIV populations, but this remains understudied. Our analysis of nearly 900 subgenomic proviral sequences from blood and lung of individuals receiving long-term HIV therapy revealed substantial within-host heterogeneity, yet some common patterns. Identical sequences (consistent with clonal expansion of infected cells) were observed in everyone, though at varying (8–86%) frequencies. The same sequence was often recovered from both blood and lung (77% of participants). Only a subset of individuals exhibited blood-lung genetic compartmentalization, and only modestly so, where this compartmentalization was sometimes due to differential frequencies of identical sequences, not the presence of truly genetically distinctive populations, across sites. Blood proviral diversity mirrored that in lung, indicating that strategies to limit the former (*e.g.* early ART) should also limit the latter. Results also revealed evidence that blood and lung proviruses can persist for >20 years within-host, and that viruses emerging in blood during treatment interruptions can re-seed both lung and blood reservoirs.

## Introduction

HIV replicates rapidly during untreated infection [1, 2], during which time the virus disseminates throughout multiple tissues [3]. Representatives of these replicating viral populations are continually archived into the reservoir, where they persist long-term, even during suppressive antiretroviral therapy (ART) [4]. Much of what we know about the HIV reservoir comes from studying proviruses that persist in blood during ART [5, 6]. In general, on-ART blood proviral diversity reflects the extent of within-host HIV evolution during untreated infection [7–9], though proviruses archived near the time of ART initiation are often overrepresented [10–12]. Clonally-expanded infected cell populations also dominate some individuals' reservoirs [13–15]. HIV persistence in tissues however is less well understood. While some studies have revealed similar proviral landscapes in blood and tissue [16–19], others support distinct viral populations in certain sites, in particular the central nervous system [3, 20].

One understudied site of HIV persistence is the lung, even though various lines of evidence identify it as a potentially distinct anatomical reservoir [21, 22]. In addition to CD4+ T-cells, HIV can persist within alveolar macrophages [23, 24], which are abundant in the lung [25]. These cells are long-lived [26], relatively resistant to both apoptosis [27–29] and cytotoxic CD8+ T-cells [30], and may harbour a higher proviral burden than blood cells [23, 31]. Unique tissue environments can facilitate the evolution of genetically distinct viral populations during untreated HIV infection, for example if founder viruses entering these environments subsequently evolve 'locally' due to restricted gene exchange with other anatomical areas [32]. Indeed, early studies uncovered differences in antiretroviral drug resistance profiles [33] and evidence of genetically compartmentalized HIV populations in blood and lung [34, 35]. These studies however were not undertaken in context of suppressive ART.

Relatively few studies have investigated lung proviral diversity and compartmentalization during ART. A recent study of autopsy tissues from ART-suppressed individuals revealed genetically diverse lung proviruses in the three participants for whom this tissue was sampled [36], though no formal compartmentalization tests were applied. Another study of 18 individuals reported limited blood-lung compartmentalization, but >80% of the participants were viremic at sample collection, so proviruses may have represented actively-replicating rather than archived viral strains [37]. A recent study of multiple autopsy tissues from long-term ART-treated individuals revealed widespread clonally-expanded cell populations and limited overall compartmentalization, but lung sequences were recovered from only two participants [38]. Another study analyzed *env* and *nef* sequences from autopsy tissues of five ART-suppressed individuals, three for whom lung sequences were obtained, but blood was not analyzed [39].

Here, we isolated nearly 900 subgenomic single-genome proviral sequences from blood and broncheoalveolar lavage (BAL) from nine individuals on long-term ART. Our goal was to characterize the diversity of proviral populations persisting in lung during ART, and to assess whether these tend to be genetically compartmentalized from those in blood, and if so in what way. That is, we asked whether blood and lung typically harbour truly genetically distinctive proviral populations (which, if so, would likely have been established prior to ART, since no or negligible viral replication occurs during this period [6, 40]), or whether these sites tend to differ only in terms of the number and distribution of clonally-expanded reservoir cells [41, 42], or whether there is generally limited evidence to support blood-lung compartmentalization of any kind. For two participants, we additionally interpreted on-ART proviral diversity in context of HIV RNA populations that replicated in plasma up to 18 years prior to proviral sampling, allowing some insights into within-host proviral dynamics.

## Results

### Within-host HIV sequence characterization

At the time of blood and lung sampling, all nine participants had been virologically suppressed on ART for a median of 9 (IQR 4–15) years (Table 1). In total, we isolated 1,043 subgenomic proviral sequences (*nef* region) by single-genome amplification from blood (N = 806) and lung (N = 237). After removal of 139 defective or hypermutated sequences and six putative within-host recombinants, 898 *nef*-intact proviral sequences remained: 715 (median 78; IQR 62–90 per participant) from blood and 183 (median 14; IQR 6–37 per participant) from lung. Participants 3, 5, and 7 yielded 10-fold fewer lung than blood proviral sequences, which limits our ability to detect compartmentalization and possibly other within-host proviral genetic features. As such, all cohort-level analyses are performed both including and excluding these participants. For participants 4 and 6, we isolated an additional 91 and 211 HIV RNA *nef* sequences respectively, by single-genome amplification from historic plasma samples collected up to 18 years prior to proviral sampling.

Overall proviral genetic diversity and anatomical distribution of *nef*-identical sequences differed markedly by participant (Table 1 and Fig 1). Participant 1 for example continued to frequently yield distinct sequences even after >100 proviruses had been collected, indicating that we did not capture the full extent of their diversity (Fig 1A). By contrast, more than 50% of participant 6's sequences were *nef*-identical. Overall, there was no consistent pattern in the frequency of distinct *nef* sequences recovered from blood versus lung (Wilcoxon matched-pairs signed rank test p>0.99 for whole cohort; p = 0.31 excluding participants 3, 5 and 7; Fig 1B). This suggests that one compartment is not inherently more likely to harbour clonally-expanded populations than the other, though it is interesting that the lungs of two participants

**Table 1. Participant information and HIV sampling details.**

| ID[a] | Sex | Duration of uncontrolled infection (years) | Years of ART prior to proviral sampling | *nef*-intact proviral lung sequences Total N (distinct N; %) | *nef*-intact proviral blood sequences Total N (distinct N; %) | pre-ART plasma HIV RNA *nef*-intact sequences[a] Total N (distinct N; %) |
|---|---|---|---|---|---|---|
| 1 | M | Unknown | 8 | 29 (29; 100%) | 88 (64; 72%) | - |
| 2 | M | >9 | 17 | 31 (13; 42%) | 60 (50; 83%) | - |
| 3* | M | >19 | <1 | 6 (6; 100%) | 75 (60; 80%) | - |
| 4 | M | >13 | 18 | 42 (7; 17%) | 78 (53; 68%) | 91 (65; 71%) |
| 5* | M | Unknown | 5 | 6 (6; 100%) | 91 (35; 38%) | - |
| 6 | M | >15 | 12 | 43 (18; 42%) | 126 (56; 44%) | 211 (99; 47%) |
| 7* | M | Unknown | >10 | 4 (4; 100%) | 79 (57; 72%) | - |
| 8 | F | Unknown | 9 | 14 (2; 14%) | 63 (58; 92%) | - |
| 9 | M | <1 | 3 | 8 (7; 88%) | 55 (40; 73%) | - |
| **Total** | | | | **183 (92; 50%)** | **715 (473; 66%)** | **302 (164; 54%)** |

[a]Participants 3, 5, and 7 yielded 10-fold fewer lung than blood proviral sequences. Within-host phylogenies for these participants are shown in S1 and S2 Figs), and all cohort-level analyses are performed both including and excluding these participants.

[b]Plasma HIV sequences were collected from three historic time points, including pre-ART, for each of participants 4 and 6. Historic plasma was not available for the other participants.

(4 and 8) were nearly entirely comprised of *nef*-identical sequences, while no such phenomenon was observed in the blood of any participant. A phylogenetic tree inferred from all *nef*-intact HIV sequences confirmed that each participant formed a monophyletic clade with a branch support value of 100%, and all participants harboured HIV subtype B (Fig 1C). Overall, participant 3 exhibited the highest within-host HIV diversity, calculated as the mean patristic (phylogenetic tip-to-tip) distance between all pairs of distinct sequences, whereas participant 9 exhibited the least diversity (Fig 1C, inset). Although clinical histories are not available for all participants, these observations are consistent with participant 3 having had HIV for at least 19 years prior to suppressing viremia on ART, where blood and lung sampling occurred only a year following viremia suppression, whereas participant 9 initiated ART shortly following diagnosis (early ART limits HIV reservoir diversity [43, 44]).

## Proviral diversity and compartmentalization in blood and lung

We next investigated proviral diversity and compartmentalization in blood and lung. To overcome the inherent uncertainty in phylogenetic inference, we inferred 7,500 phylogenies per participant using Bayesian approaches. For each participant, the highest likelihood phylogeny from among these 7,500 reconstructions is shown in the main figure, and the consensus trees are shown in S3, S4, S5, S6, and S7 Figs. As is standard practise, we applied more than one type of compartmentalization test, as these assess different genetic features of the populations in question ([45] and see below). Specifically, we applied two genetic distance-based and two tree-based tests, and conservatively required a dataset to return a statistically significant result (p<0.05) on at least one test *per type*, in order to be declared compartmentalized.

The genetic distance-based tests were Analysis of Molecular Variance (AMOVA) [46], and Hudson, Boos and Kaplan's nonparametric test for population structure ($K_{ST}$) [47]. AMOVA calculates an association based on the genetic diversity of sequences within and between compartments, where variability is calculated from the sum of the squared genetic distances between sequences. $K_{ST}$ compares the mean pairwise distances between sequences from different versus the same compartment. Compartmentalization is supported if the latter are smaller

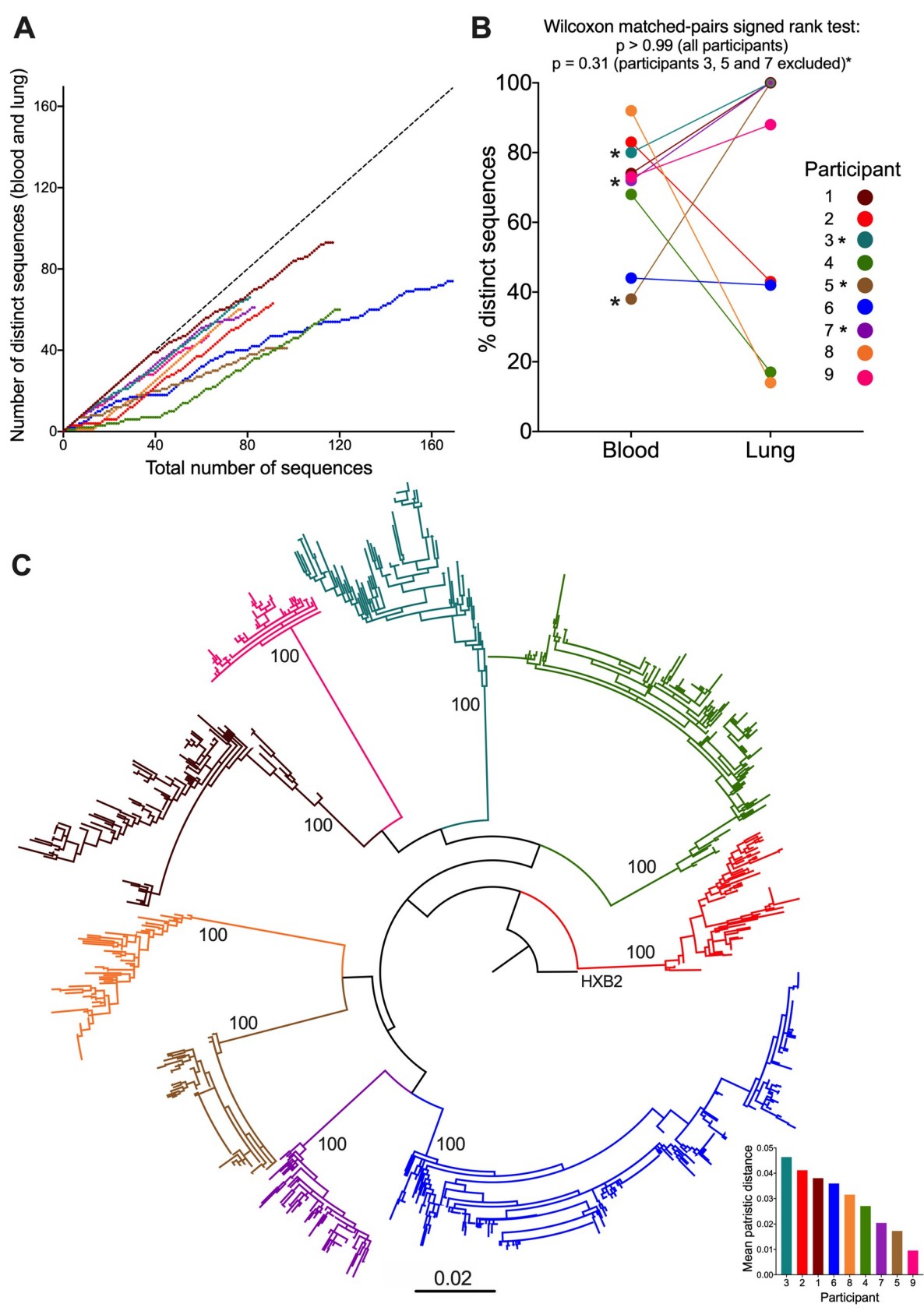

**Fig 1. Within-host HIV sequence characterization.** (A) The number of *distinct nef* proviral sequences collected from blood and lung, as a function of the total number of sequences collected for each participant. This helps estimate the extent to which sampled sequences captured within-host diversity. The dotted diagonal shows what the relationship would look like if every sampled sequence was distinct. (B) Proportion of distinct proviral sequences isolated from blood and lung of each participant. The p-values were calculated across all participants, and excluding participants 3, 5, and 7 (marked by asterisks), using the Wilcoxon matched-pairs signed rank test. (C) Maximum-likelihood phylogeny inferred from 898 proviral and 302 plasma HIV RNA sequences, rooted on the HIV subtype B reference strain HXB2. Numbers on internal branches indicate bootstrap values. Scale in estimated substitutions per nucleotide site. Inset: participants ranked from highest to lowest genetic diversity (calculated as mean patristic distances of distinct sequences).

than the former, with statistical significance derived via a population-structure randomization test. Both test statistics (phi [$\phi$] and $K_{ST}$, respectively) range from 0 (no compartmentalization) to 1 (complete compartmentalization), where p-values are derived from 1,000 permutations (Table 2).

The two tree-based methods were the Slatkin-Maddison (SM) [48] and Correlation Coefficient (CC) tests [49]. SM determines the minimum number of migrations between compartments to explain the distribution of compartments at the tree tips: the smaller the number of migrations, the stronger the compartmentalization support. Statistical support is based on the number of migrations that would be expected in a randomly-structured population, simulated by permuting compartment labels between sequences. The CC test correlates the phylogenetic distances between two sequences (defined here as the number of internal branches separating them in the tree) with information on their compartment of isolation. Coefficients can range from -1 to +1, where larger positive values denote stronger support for compartmentalization, and values closer to zero (and negative values) denote no compartmentalization. Statistical support is derived from estimating the distribution of these coefficients by permuting compartment labels between sequences. The tree-based tests were applied to all 7,500 phylogenies per participant and results were averaged, such that a mean p<0.05 was considered statistically significant. Finally, each dataset was analyzed two ways: restricting to only distinct *nef* sequences per compartment ("distinct" analysis), and including all sequences ("overall" analysis). This is because identical sequences, especially when differentially abundant across compartments, increase the likelihood of obtaining a significant p-value, but such results are usually attributable to differential clonal expansion, rather than divergent HIV evolution between compartments [37, 42].

Only two participants, 2 and 6, demonstrated significant evidence of blood-lung proviral compartmentalization when only distinct *nef* sequences were analyzed (Fig 2). From participant 2, we recovered 60 sequences from blood (83% distinct in this compartment) and 31 from lung (42% distinct in this compartment); the phylogeny and adjacent highlighter plot illustrate within-host genetic and amino acid diversity (Fig 2A). Four *nef* sequences were found in both blood and lung; for three of these, we recovered two or more copies of this sequence in one or both compartments. The most abundant sequence was observed 12 times (all in lung). When analyzing only distinct sequences per compartment, participant 2's dataset met our criteria for compartmentalization, with both AMOVA ($\phi$ = 0.08, p = 0.016) and CC (coefficient = 0.11, p = 0.034) returning statistically significant results (Fig 2B and Table 2). In both cases however the test statistic value was low, indicating that the extent of compartmentalization was modest. When all sequences were considered, all four tests returned significant results for this participant (with AMOVA and $K_{ST}$ yielding p = 0, SM p = 0.046, and CC p = 0.037), though the overall magnitude of compartmentalization remained modest (*e.g.* though AMOVA $\phi$ increased from 0.08 to 0.23, the CC coefficient decreased from 0.11 to 0.08).

Participant 6 was distinctive in that they harboured the highest overall frequency of *nef*-identical sequences (only 44% of the 126 sequences recovered from blood and 42% of those from lung were distinct in those compartments) and that these tended to be found in a single

**Table 2. Blood-lung compartmentalization results.**

| Dataset | Test type | Result[a] | 1 | 2 | 3 | 4 | 5 | 6 | 7 | 8 | 9 |
|---|---|---|---|---|---|---|---|---|---|---|---|
| | | | | | | **Participant** | | | | | |
| **Distinct** | *Distance* | AMOVA | 0.01 | 0.08 | 0 | 0 | 0.005 | 0.17 | 0 | 0 | 0 |
| | | p-value | 0.16 | **0.016** | 0.77 | 0.46 | 0.39 | **0.011** | 0.87 | 0.42 | 0.77 |
| | | $K_{ST}$ | 0 | 0.04 | 0.02 | 0.02 | 0.007 | 0.04 | 0.01 | 0.02 | 0.005 |
| | | p-value | 0.97 | 0.07 | 0.4 | 0.37 | 0.78 | **0.024** | 0.67 | 0.55 | 0.9 |
| | *Tree* | SM | 24 | 10 | 5 | 7 | 5 | 15 | 4 | 2 | 6 |
| | | p-value | 0.5 | 0.16 | 0.7 | 0.99 | 0.63 | 0.37 | 0.99 | 0.99 | 0.99 |
| | | CC | -0.01 | 0.11 | 0.04 | 0.08 | 0.07 | 0.09 | -0.02 | 0.12 | 0.05 |
| | | p-value | 0.62 | **0.034** | 0.26 | 0.09 | 0.21 | **0.039** | 0.63 | 0.05 | 0.23 |
| **Overall** | *Distance* | AMOVA | 0.006 | 0.23 | 0.04 | 0.07 | 0.11 | 0.49 | 0 | 0.44 | 0 |
| | | p-value | 0.27 | **0** | 0.18 | **0** | 0.06 | **0** | 0.85 | **0** | 0.73 |
| | | $K_{ST}$ | 0 | 0.08 | 0.01 | 0 | 0.03 | 0.2 | 0.01 | 0.06 | 0.01 |
| | | p-value | 0.97 | **0** | 0.35 | 0.97 | 0.1 | **0** | 0.5 | **0.002** | 0.47 |
| | *Tree* | SM | 24 | 14 | 5 | 13 | 5 | 18 | 4 | 2 | 7 |
| | | p-value | 0.2 | **0.046** | 0.68 | **0** | 0.56 | **0** | 0.99 | **0** | 0.89 |
| | | CC | -0.04 | 0.08 | 0.01 | 0.15 | 0.14 | 0.3 | -0.02 | 0.23 | 0.01 |
| | | p-value | 0.87 | **0.037** | 0.42 | **0.012** | **0.037** | **0** | 0.64 | **0.002** | 0.38 |

[a]For each test, the first row reports the test statistic or output. For AMOVA, this is the $\phi$ statistic, and for $K_{ST}$, this is the $K_{ST}$ statistic. For both of these, 0 indicates no compartmentalization and 1 indicates complete compartmentalization. For the Slatkin-Maddison (SM) test, the mean number of inferred migrations between compartments (averaged over all 7,500 trees) is reported. The smaller this number relative to total and per-compartment dataset size, the stronger the support for compartmentalization. For the correlation coefficient (CC) test, which uses Pearson's correlation, the mean correlation coefficient (averaged over all 7,500 trees) is reported. Coefficients can range from -1 to +1, where larger positive values denote stronger support for compartmentalization, and values closer to zero (and negative values) denote no compartmentalization. For the SM and CC tests, the mean p-value from all 7,500 trees is reported. Statistically significant p-values are shown in bold.

compartment (Fig 2C). The most abundant sequence was observed 38 times (all in blood). An additional eight *nef* sequences were observed five or more times, most often in a single compartment, though there were four instances where a sequence was observed in both blood and lung. Like participant 2, participant 6's dataset exhibited modest yet statistically significant evidence of compartmentalization when considering only distinct *nef* sequences per compartment (AMOVA $\phi$ = 0.17, p = 0.011; $K_{ST}$ = 0.04, p = 0.024; CC coefficient = 0.09, p = 0.039). Support for compartmentalization became more pronounced when all sequences were considered (*e.g.* with AMOVA $\phi$ reaching 0.49, and all tests returning p = 0) (Fig 2D; Table 2).

Two additional participants, 4 and 8, showed evidence of compartmentalization only when all sequences were considered (Fig 3). Participant 4 was remarkable in that they harboured a single *nef* sequence that was recovered 42 times (36 times in lung, 7 in blood) (Fig 3A). An additional four sequences were shared across compartments, but otherwise the remaining blood and lung sequences were relatively diverse. When considering all sequences, three of the four compartmentalization tests returned significant results (AMOVA and SM returned p = 0; CC returned p = 0.012), though the magnitude of this compartmentalization was again relatively modest (*e.g.* AMOVA $\phi$ was 0.07) (Fig 3B, Table 2). Participant 8 was remarkable in that 13 out of 14 sequences isolated from lung were identical and distinct to this compartment, whereas blood sequences were relatively diverse with few duplicates (Fig 3C). All four tests returned significant evidence of compartmentalization when considering all sequences (with AMOVA and SM both returning p = 0, and $K_{ST}$ and CC both returning p = 0.002), with some tests indicating a stronger compartmentalization magnitude (*e.g.* AMOVA $\phi$ was 0.44) (Fig 3D, Table 2).

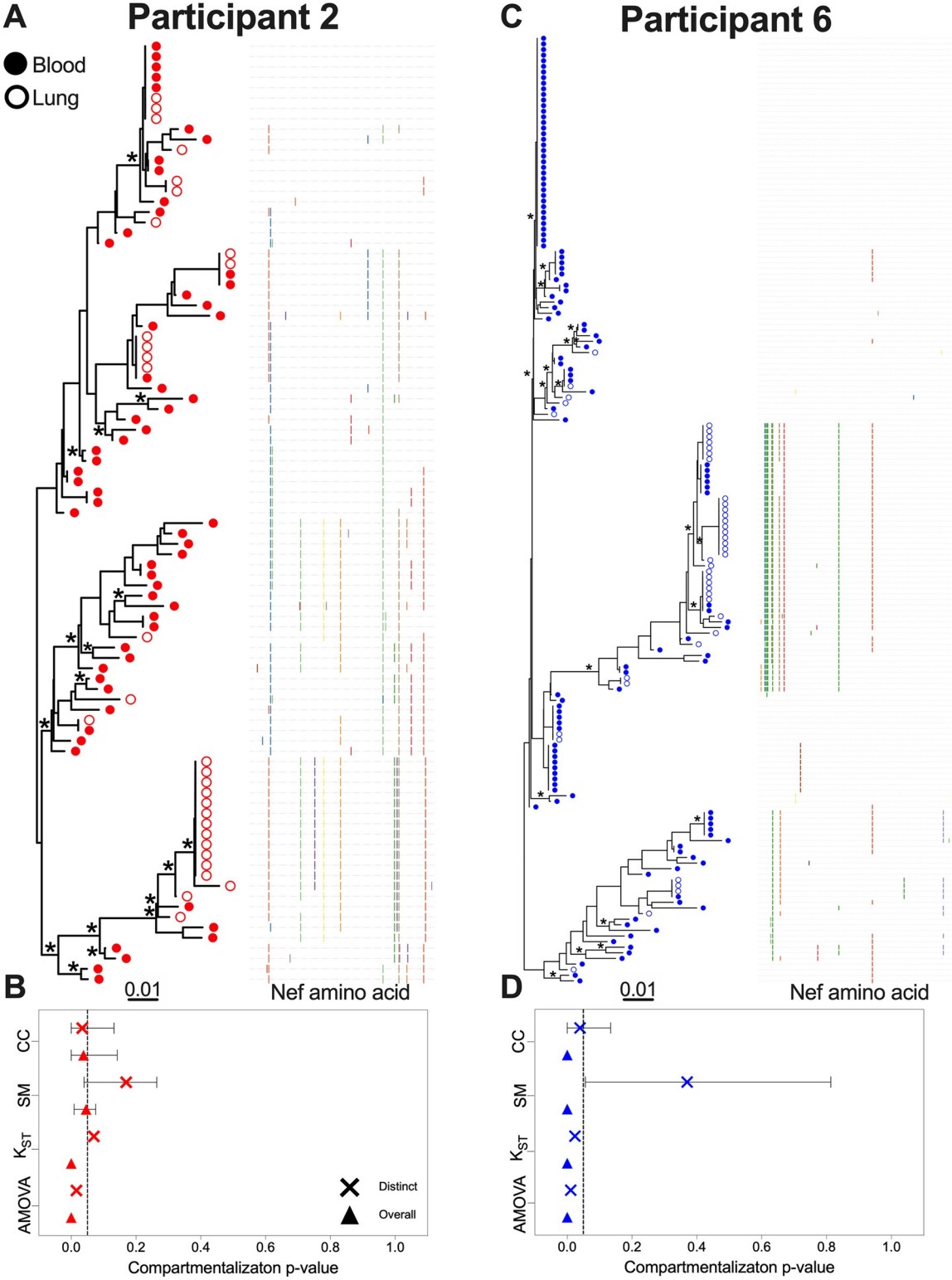

**Fig 2. Within-host phylogenies and compartmentalization results for participants 2 and 6.** Each participant's top panel shows the highest likelihood phylogeny derived from Bayesian inference, rooted at the midpoint. Filled circles denote blood sequences; open circles denote lung sequences. Asterisks identify nodes supported by posterior probabilities ≥70%. The adjacent plot shows amino acid diversity of each sequence, with coloured ticks denoting non-synonymous substitutions with respect to the sequence at the top of the phylogeny. The bottom panel shows p-values of the four tests of genetic compartmentalization between blood and lung

(AMOVA = Analysis of Molecular Variance; $K_{ST}$ = Hudson, Boos and Kaplan's nonparametric test for population structure; SM = Slatkin-Maddison test; CC = Correlation Coefficient test). The "×" symbol shows the p-value based on distinct sequences per compartment, and the triangle shows the p-value based on overall sequences. For SM and CC tests, the bars represent the 95% HPD interval of the p-value derived from all 7,500 trees.

The remaining participants showed no evidence of blood-lung genetic compartmentalization of any kind (Figs 4, S1 and S2). From participant 1, we recovered 88 sequences from blood (72% distinct in this compartment) and 29 from lung (100% distinct in this compartment), where seven sequences were found in both compartments (Fig 4A). The most abundant sequence was observed six times (five in blood, one in lung). No evidence of compartmentalization was detected by any test regardless of the inclusion of identical sequences (Fig 4B). From participant 9 we recovered 55 sequences from blood (73% distinct in this compartment) and eight from lung (of which 7 were distinct and relatively divergent from one another) (Fig 4C). Two sequences, including the one duplicate *nef* sequence recovered from lung, were found in both compartments. No compartmentalization was detected by any test regardless of the inclusion of identical sequences (Fig 4D). For the remaining three participants (3, 5 and 7) our power to detect compartmentalization was limited due to low sequence recovery from lung. Nevertheless, in all three participants, all recovered lung sequences were distinct and relatively divergent from one another, indicating that the proviral pool in lung was genetically diverse (S1 and S2 Figs). Participant 5 was also remarkable for the recovery of 41 identical sequences, all in blood, consistent with clonal expansion (S1 Fig). Overall compartmentalization results, summarized by participant and degree of statistical significance, are shown in Fig 5A.

We also investigated the relationship between blood and lung proviral genetic diversity, calculated two ways. First, we computed the mean patristic (tip-to-tip) distances separating blood sequences from one another, and lung sequences from one another, across all 7,500 phylogenies per participant, and averaged these values across all trees (Fig 5B). Correlating these values yielded a Spearman's $\rho = 0.98$, p<0.0001 when all participants were considered, and a Spearman's $\rho = 0.94$, p = 0.01 when participants 3, 5 and 7 were excluded. Second, we computed the overall mean phylogenetic diversity per compartment (as the sum of edge lengths of distinct sequences per compartment) averaged over all trees (Fig 5C). Correlating these values yielded a Spearman's $\rho = 0.76$, p = 0.02 between blood and lung when all participants were considered, and a Spearman's $\rho = 0.60$, p = 0.24 when participants 3, 5 and 7 were excluded. Despite limited recovery of lung sequences in some participants, this nevertheless provides strong evidence that blood proviral diversity on ART generally reflects that in lung.

## Exploring proviral diversity in context of historic within-host plasma HIV RNA populations

For participants 4 and 6, plasma samples collected pre-ART up to 18 years prior to proviral sampling, and during historic viremia rebound events, were available. This allowed us to interpret on-ART proviral diversity in context of pre-ART HIV populations, and to investigate evolutionary relationships between rebound viruses and the persistent proviral pool. Interpreting on-ART proviral sequences in this context requires rooting the tree at a location that represents the most recent common ancestor of the collected sequences. If a dataset contained sequences dating back to infection, a correctly-placed root would represent the transmission event. If the dataset did not contain sequences dating all the way back to infection, a correctly placed root would represent a within-host descendant of the transmitted/founder virus that arose later during infection. Either way, rooting the tree allows the sampled sequences to be temporally ordered, where those closer to the root can be inferred as being older, whereas

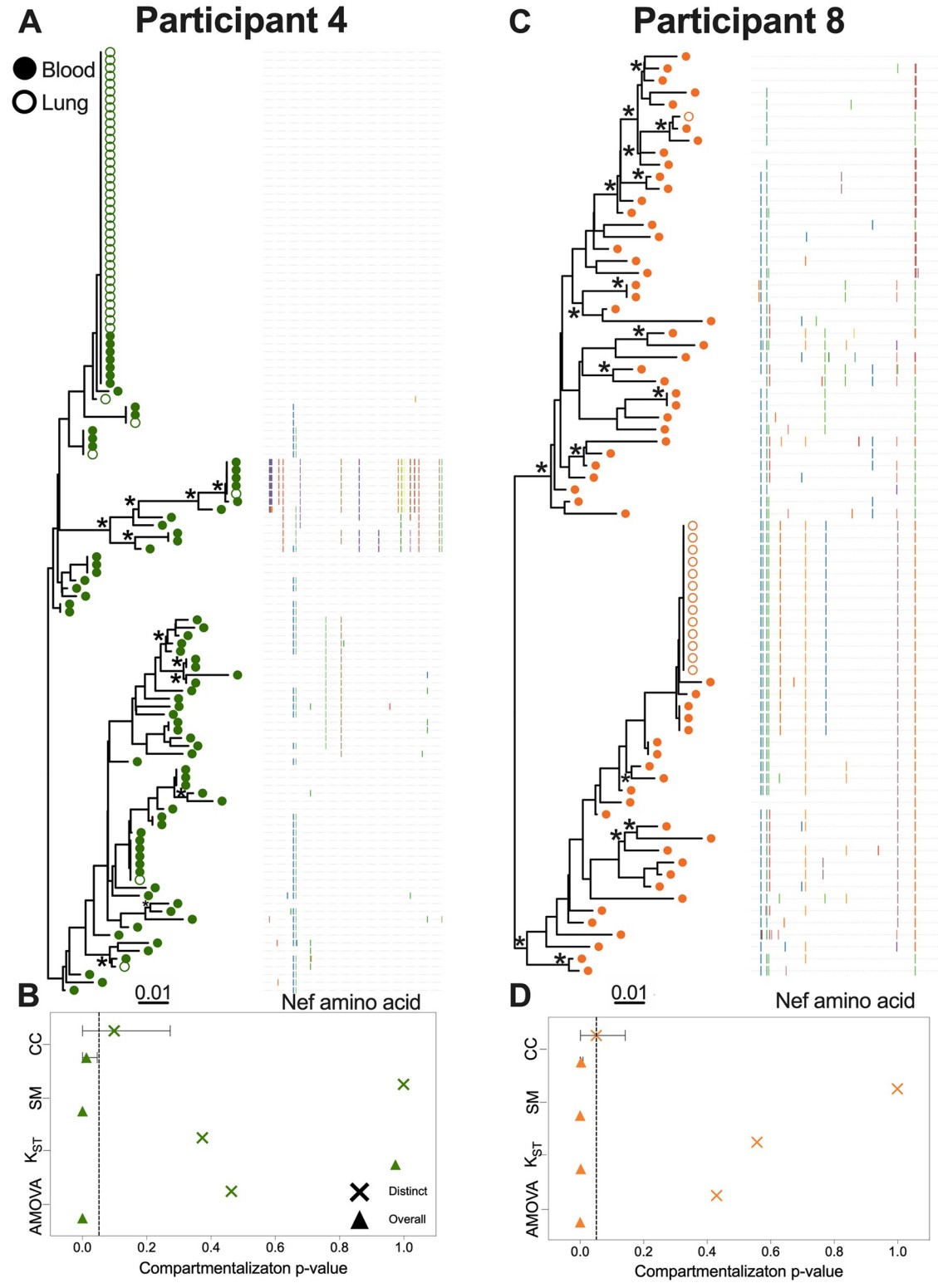

**Fig 3. Within-host phylogenies and compartmentalization results for participants 4 and 8.** Legend as in Fig 2.

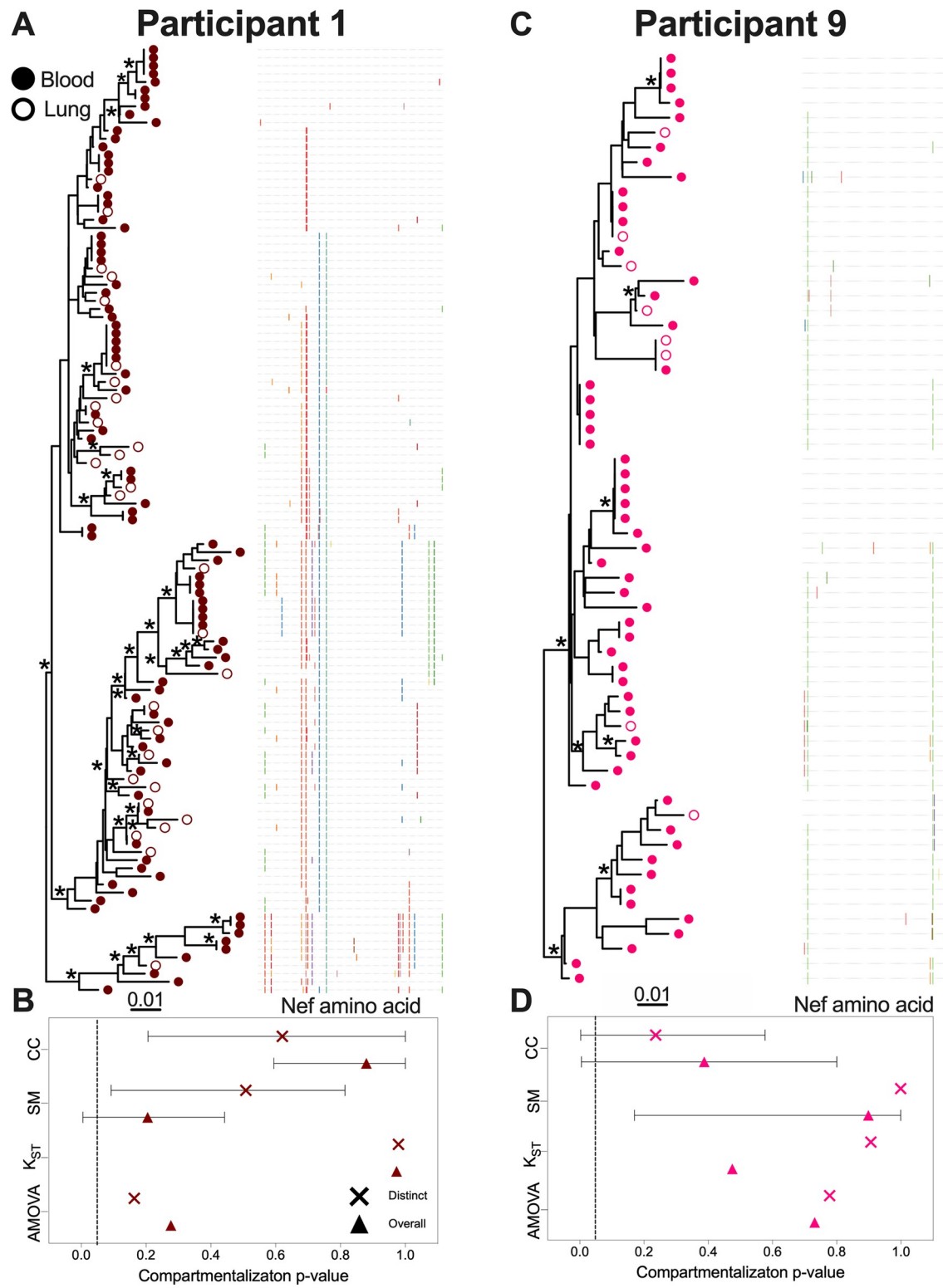

**Fig 4. Within-host phylogenies and compartmentalization results for participants 1 and 9.** Legend as in Fig 2.

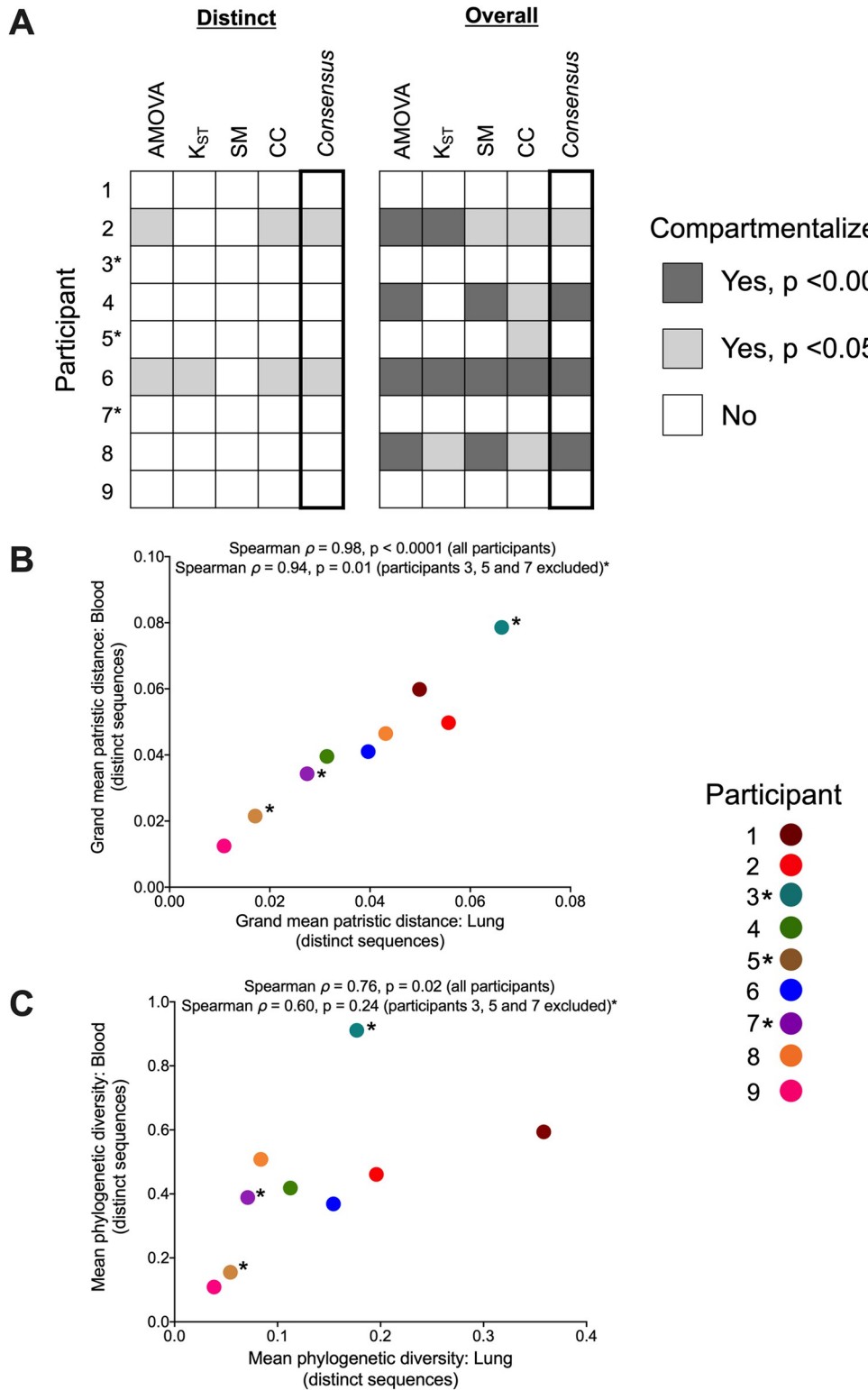

**Fig 5. Summary of genetic compartmentalization results and blood-lung proviral diversity.** (A) Summary of genetic compartmentalization results for the four tests used, along with the "consensus" result, where a dataset is declared compartmentalized if at least one test per type (genetic distance vs. tree-based) gave a statistically significant result. Dark grey squares denote compartmentalization with p<0.001, light grey squares denote compartmentalization with p<0.05; white squares denote no compartmentalization. The "Distinct" panel summarizes compartmentalization

results when limiting analysis to distinct sequences per compartment; the "Overall" panel summarizes results when all sequences are included. (B) Spearman's correlation between blood and lung proviral diversity, where values represent the grand mean of within-host patristic distances separating all pairs of distinct sequences per compartment, averaged over all 7,500 trees per participant. Spearman's correlation and p-values were computed across all participants and after excluding participants 3, 5, and 7 (shown by asterisks). (C) Spearman's correlation between blood and lung proviral diversity, where values represent mean phylogenetic diversity of distinct sequences per compartment, computed from all 7,500 trees per participant.

those most divergent from the root are younger. One method to root within-host phylogenies is by leveraging information from plasma HIV RNA sequences sampled longitudinally pre-ART [7, 50, 51], but this was not possible here because only a single pre-ART plasma sample was available per participant. Instead, we used another established method, an outgroup, to root the trees [7], and then used a topological test to support root placement (see below and methods).

Participant 4 was diagnosed with HIV in January 1987 but initiated ART only in February 2000 (Fig 6A). Viremia was generally suppressed thereafter except for brief rebound episodes in 2004 and 2005 following ART interruption. We isolated 36, 33 and 22 intact HIV RNA *nef* sequences, respectively, from plasma sampled in January 2000 (pre-ART) and the 2004 and 2005 viremic episodes, inferred 7,500 phylogenies from these along with the 120 proviral sequences isolated from blood and lung during ART, and rooted these using an outgroup (representative tree in Fig 6B; consensus tree in S8A Fig). Notably, the proviruses sampled after almost 18 years on ART interspersed throughout the whole phylogeny and represented all of the sequences near the root, which was far deeper in the tree than any sampled plasma HIV RNA sequence. This suggests that by far the most ancestral sequences sampled from this participant were on-ART proviruses. To further support this root placement, we compared an unconstrained maximum-likelihood phylogeny to one where the sequences from the earliest plasma sampling (in February 2000) were constrained to be separated from the sequences sampled from plasma in 2004 and 2005, but not necessarily from the proviral sequences. The unconstrained tree's log-likelihood was 44 units higher than the constrained one, yielding p-values < 0.05 in support of the unconstrained trees on all five topology tests performed [52–56]. This suggests that the plasma sequences sampled in Feb 2000, despite being sampled the earliest in calendar time, do not constitute an ancestral lineage that gave rise to the remainder of the participants' sequences, thus supporting the outgroup-guided root placement.

Participant 4's proviruses' wide-ranging root-to-tip distances, and the enrichment of ancestrally-placed sequences in this group, are also apparent in Fig 6C (*e.g.* lung mean proviral divergence from the root was 0.049 substitutions per nucleotide site compared to 0.081 among sequences circulating in plasma just prior to ART). Proviruses persisting in blood during ART were also on average more diverse (mean 0.052 substitutions/site) than any plasma populations sampled previously (Fig 6D). The 2005 rebound plasma sequences were particularly low-diversity (mean 0.025 substitutions/site), suggesting that only a genetically restricted subset of replication-competent viruses reactivated at that time.

Overall, these observations suggest that, while participant 4's *replication-competent* reservoir largely dates to around the time of ART initiation (as evidenced by the similar root-to-tip divergence measurements of the pre-ART sequences circulating in 2000 to those that rebounded in plasma in both 2004 and 2005), their overall proviral pool, that would also include defective proviruses, features sequences that are far more ancestral than this (as evidenced by the abundance of proviral sequences close to the root and the complete lack of plasma sequences within these ancestral subclades). The data also suggest that the 2004 and 2005 viremia rebound events re-seeded the reservoir to some extent. This is evidenced by four instances where we recovered blood and/or lung proviruses in 2018 that were genetically

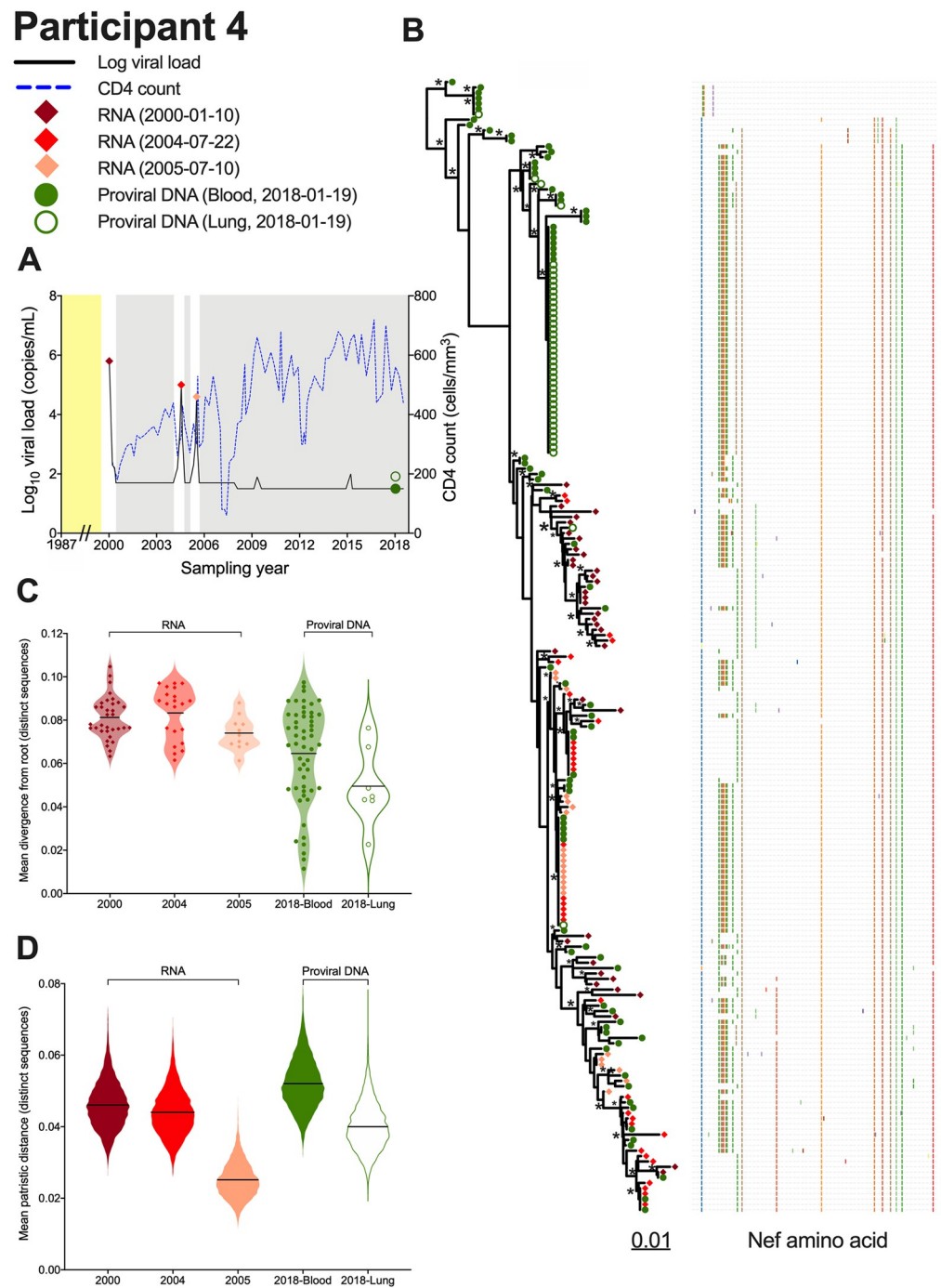

**Fig 6. Interpreting proviral diversity in context of prior within-host HIV evolution: participant 4.** (A) Plasma viral load (solid black line), CD4+ T-cell count (dashed blue line) and samples analyzed (colored diamonds and circles). Yellow shading denotes period following infection where no clinical information is available. Grey shading denotes periods of suppressive ART. (B) Highest likelihood phylogeny derived from Bayesian inference, outgroup rooted. Scale shows estimated substitutions per nucleotide site. The adjacent plot shows amino acid diversity, with coloured ticks denoting non-synonymous substitutions with respect to the reference sequence at the

top of the tree. Asterisks identify nodes supported by posterior probabilities ≥70%. (C) Mean divergence from the root of each distinct sequence, averaged over all 7,500 trees, stratified by sampling date and sample type. The black line represents the grand mean. (D) Within-host HIV sequence diversity, calculated as the mean patristic distance between all distinct sequences collected at that time point for that sample, where the values for each of the 7,500 trees are represented as a violin plot. The black line represents the grand mean.

identical to 2004 and/or 2005 plasma rebound viruses, where in two of these instances the sequence was recovered multiple times in both plasma HIV RNA and proviral forms. By contrast, we isolated only one provirus that was genetically identical to a sequence that circulated in plasma just prior to ART initiation (Fig 6B, near the bottom of the tree).

Participant 6 was diagnosed with HIV in June 1982 and initiated ART in 1997 (Fig 7A). Treatment was interrupted from February to April 2002, and again from February 2006 to August 2009, during which time pVL rebounded to $\geq 4 \log_{10}$. There was another brief viremic episode in 2012 (average 215 RNA copies/mL). A pre-ART plasma sample from October 1996, along with samples from 2006 and 2009 during the second treatment interruption, were available. We isolated 211 intact plasma HIV RNA *nef*-intact sequences (68 from 1996; 86 from 2006; 57 from 2009), inferred 7,500 phylogenies from these along with the 169 proviral sequences isolated from blood and lung during ART, and outgroup rooted these trees (representative tree in Fig 7B; consensus tree in S8B Fig). Similar to participant 4, the sequences closest to the root were on-ART proviral sequences sampled in 2014 (from both blood and lung), not the pre-ART plasma sequences sampled in 1996, consistent with the proviral pool containing more ancestral sequences than those circulating in plasma immediately prior to ART. In further support of this root placement, an unconstrained phylogeny returned a log-likelihood of 64 units higher than one where the 1996 plasma sequences were constrained to be separated from the 2006–2009 ones (all p<0.05), suggesting that that the 1996 plasma sequences do not constitute an ancestral lineage that gave rise to the remainder of the participants' sequences.

The wide-ranging root-to-tip distances of participant 6's on-ART proviruses, and the enrichment of ancestrally-placed sequences in this group, is also apparent in Fig 7C. Unlike participant 4 however, participant 6's root location was not too much deeper than their earliest sampled plasma HIV RNA sequences. This suggests that the oldest sampled proviruses did not circulate too much earlier than the pre-ART sampling date of October 1996. Similar to participant 4, the plasma HIV RNA sequences that circulated in 1996 just prior to ART initiation ranged widely in terms of root-to-tip divergence (grand mean of 0.067 substitutions/site; Fig 7C). They were also the most diverse of any time point sampled (mean 0.059 substitutions/site; Fig 7D), which is not unexpected given that these sequences represented the end products of >15 years of uncontrolled within-host HIV evolution. In contrast, the sequences that rebounded in plasma in 2006 were far less diverse than the 1996 pre-ART sequences (grand mean diversity 0.033 substitutions/site; Fig 7D). No plasma rebound viruses for example were found in the subclade near the top of the tree that was exclusively comprised of pre-ART plasma and on-ART proviral sequences. This suggests that the replication-competent HIV variants that rebounded in plasma in 2006 were a restricted subset of the viral population that circulated just prior to ART initiation. These plasma rebound variants evolved over the next three years, as shown by the increased root-to-tip divergence and diversity of the 2009 plasma sequences compared to the 2006 ones. Like participant 4, participant 6's phylogeny also suggests that these rebound plasma viruses re-seeded the reservoir. This is evidenced by four instances where we recovered blood and/or lung proviruses in 2014 that were genetically identical to sequences isolated from plasma during ART interruption, including 3 instances of proviruses identical to 2009 plasma sequences (near bottom of tree) and one instance of a provirus identical to a sequence that was isolated 52 times from plasma in 2006 (middle of tree). By contrast, we did not isolate any proviruses that were genetically identical to any plasma viruses that

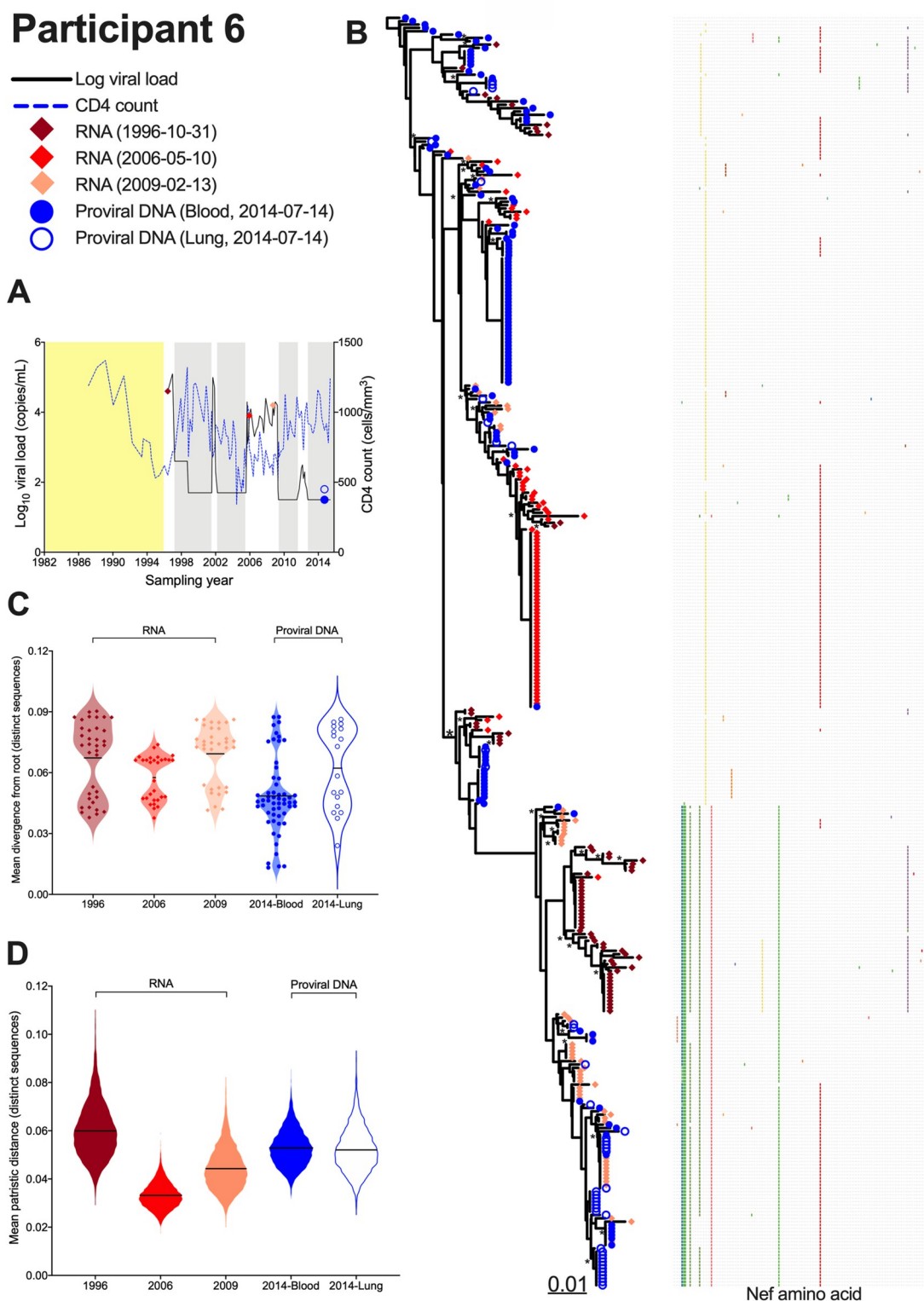

**Fig 7. Interpreting proviral diversity in context of prior within-host HIV evolution: participant 6.** (A) Plasma viral load (solid black line), CD4+ T-cell count (dashed blue line) and samples analyzed (colored diamonds and circles). Yellow shading denotes period following infection where clinical information is incomplete. Grey shading denotes periods of suppressive ART as per the HIV RNA pVL assay used at the time (until 1998, values <400 copies/mL indicated undetectable viremia; later assays used a <50 copies/mL cutoff) [91]. (B) Highest likelihood phylogeny derived from Bayesian inference, outgroup rooted. Scale

in estimated substitutions per nucleotide site. The adjacent plot shows amino acid diversity, with coloured ticks denoting non-synonymous substitutions with respect to the sequence at the top of the tree. Asterisks identify nodes supported by posterior probabilities ≥70%. (C) Mean divergence from the root of each distinct sequence, averaged over all 7,500 trees, stratified by sampling date and sample type. The black line represents the grand mean. (D) Within-host HIV sequence diversity, calculated as the mean patristic distance between all distinct sequences collected at that time point for that sample, where the values for each of the 7,500 trees are represented as a violin plot. The black line represents the grand mean.

circulated in 1996 pre-ART. Also of note, the sampled proviruses were somewhat less diverse and divergent than the 1996 pre-ART sequences (Fig 7C and 7D), largely because the latter featured a distinct subclade, that was quite divergent from the root, that contained only a single sequence from one other timepoint, a 2006 plasma sequence (about two-thirds down the tree). This suggests that representatives of this clade, which were abundant in plasma just prior to ART initiation, did not persist in abundance in the reservoir.

## Discussion

Our recovery of *nef* proviral sequences from the lungs of all participants, albeit in limited numbers in some cases (in particular, participants 3, 5 and 7), confirms this organ as a site of long-term HIV persistence during ART. Our results also revealed heterogeneous within-host proviral genetic composition. The abundance of *nef*-identical sequences for example, consistent with clonally-expanded cell populations, ranged widely. Whereas a single sequence dominated in the lung in some participants (*e.g.* 4 and 8), in others a specific sequence dominated in blood (*e.g.* 5). In yet others, few or no *nef*-identical sequences were recovered (*e.g.* 1). While this supports clonal expansion of reservoir cells as a common driver of HIV persistence during ART [14, 15, 57], it indicates that we cannot make generalizations regarding where clonally-expanded reservoir cells are more likely to be found, and also reveals that large clonally-expanded populations are not abundant in everyone at all times. Notably, the extent of proviral genetic diversity in blood broadly reflected that in lung.

We also investigated the extent of blood-lung genetic compartmentalization, and whether this tends to manifest as distinct viral lineages across sites, or simply as differential proportions of clonally-expanded populations. With respect to the former, no dataset was so genetically compartmentalized that it was visually apparent in a phylogeny, but when we applied formal tests two participants (2 and 6) showed modest yet statistically significant evidence of distinct viral lineages across sites (participant 6 also showed evidence of differential clonal expansion across sites, as evidenced by the stronger test statistics obtained in the "overall" analysis). The type of compartmentalization that is solely attributable to differential clonal expansion was detected in two additional participants (4 and 8), where the magnitude was generally stronger. Compartmentalization was not detected in the remainder of participants, though for three of these (3, 5 and 7) insufficient recovery of lung sequences limited our power to detect it. Despite this, our results nevertheless suggest that "true" blood-lung proviral compartmentalization on ART (defined as the presence of genetically distinct viral variants) is not the norm, and when present it is generally of modest magnitude. The type of proviral genetic compartmentalization that is due to differential clonal expansion is more common, but still by no means universal. These observations are consistent with studies of on-ART viral diversity in lung [37, 38], female genital tract [42], testes [41], and lymph nodes [58].

Archived plasma also allowed us to study on-ART proviruses in the context of historic within-host HIV RNA diversity in two participants. In participant 4, proviral sequences sampled on ART were substantially more diverse than plasma viral populations that circulated immediately pre-ART. Proviruses sampled on ART also extended deep into the tree all the way to the root, supporting the notion that, at the time of sampling in 2018, some of these

sequences had been persisting for more than 20 years. Somewhat in contrast, participant 6's on-ART proviruses were broadly comparable to plasma HIV RNA populations circulating immediately prior to ART initiation in terms of diversity. Though the sequences closest to participant 6's root were also proviruses sampled on ART, a number of pre-ART plasma sequences also fell relatively close to the root. This suggests that, unlike participant 4, participant 6's oldest proviruses did not pre-date their ART initiation date by very long. These results are consistent with recent observations that proviruses persisting on ART are typically enriched in sequences that integrated around ART initiation [10, 12, 50], though some individuals preserve larger numbers of ancestral sequences [7, 8, 50, 51]. This can be explained, at least in part, by differential pre-ART proviral dynamics: while HIV sequences continually enter the reservoir during untreated infection, turnover is much more rapid during this period than during ART [50, 59–61]. During untreated infection, the estimated average half-life of persisting proviruses (most of which are defective [62–64]) is on the order of 1 year [11, 12, 59], which means that most early proviruses will have been eliminated by the time ART is initiated. Pre-ART proviral turnover rates also vary between individuals [50, 59], so some people will preserve a greater proportion of ancestral sequences in their reservoir than others.

For both participants 4 and 6, historic viremia rebound events that occurred after ART interruption also likely influenced their proviral composition. These rebound sequences are valuable because they represent replication-competent HIV emerging from the reservoir. Of note, participant 4's rebounding HIV RNA sequences were similar in terms of root-to-tip divergence to those that circulated immediately prior to ART, and none were located near the root. This suggests that, while participant 4's on-ART proviral pool contained a substantial proportion of older sequences, their *replication-competent* reservoir largely dates to around the time of ART initiation. The phylogenies of both participants 4 and 6 also strongly suggest that these historic viral rebound events re-seeded their reservoirs to some extent. Evidence for this comes from the repeated recovery of blood and lung proviruses many years later during suppressive ART, that were genetically identical to plasma HIV RNA sequences that were present during the rebound events.

Our study has several limitations. Sequence recovery from BAL was challenging, and, similar to published studies [38, 39], only limited numbers of lung sequences were recovered from some participants. Limited sequence recovery could be due to low proviral loads, but there was insufficient DNA to confirm this (the total number of BAL cells sampled, available for four participants, was between 2.15–13 million). Smoking-related compounds may have additionally reduced BAL HIV amplification efficiency (7 participants were current or former smokers, and BAL was visibly darkened for 5 of these), though there was no obvious correlation between smoking and lung sequence recovery (and participant 7, who yielded the lowest number of lung sequences, was a non-smoker). Regardless, low lung sequence numbers limited our ability to estimate proviral parameters (% distinct, *nef*-intact, etc) and detect compartmentalization in a subset of participants. Sequence amplification bias from limited starting material is also a theoretical possibility, but for five participants we had amplified proviruses from independent sample aliquots, and testing the two batches of data did not support compartmentalization (defined as p<0.05 on at least one distance- and tree-based test) in any case (S9 Fig). Since blood and lung proviruses were sampled at a single time point on ART, we cannot establish the mechanism of compartmentalization when it was detected. In these cases, distinctive HIV populations detected in blood and lung were likely already established prior to ART (since HIV replication is negligible or non-existent during ART [6, 40]), but we cannot exclude the possibility that proviral populations pre-ART were similar, but that certain proviral lineages subsequently decayed at differential rates between sites, yielding distinctive populations when sampled years later. Similarly, datasets that exhibited the "differential clonal expansion" type of compartmentalization are likely the result of dynamic changes in clonal expansion during

ART [13], but we cannot rule out the existence of imbalanced clonally-expanded populations prior to ART, nor can we conclusively identify the site where the clonal expansion occurred (*e. g.* clones could have arisen in one compartment, migrated to another, then largely disappeared from the former). Because we performed subgenomic sequencing, we cannot definitively classify identical sequences as being from clonally-expanded cells. Furthermore, most proviral sequences are likely not from intact HIV genomes, since most proviruses persisting during ART harbour large deletions [62–64]. In fact, even subgenomic plasma HIV RNA sequences may not necessarily be derived from intact viral genomes, as recently demonstrated [65]. Intact subgenomic sequences however are still appropriate, and routinely used, for inferring evolutionary relationships and compartmentalization testing (*e.g.* [3, 7, 9, 41, 51]). This is because large deletions arise, likely in a single step, during *in vivo* reverse transcription of HIV RNA genome (this is supported by the observation that deletions reproducibly occur at locations flanked by identical sequence motifs [63, 64, 66]). As such, intact subgenomic proviral regions still reflect the sequence of the original infecting virus and are therefore appropriate for evolutionary inference. As biological material was limited, we isolated proviruses directly from blood and BAL, so the cell types that harboured these sequences remain unknown (though BAL cell phenotyping, available for four participants, indicated these were 92–99% macrophages). Our isolation of large numbers of identical sequences from the lungs of some individuals, strongly suggestive of clonal expansion, suggests that these are derived from CD4$^+$ T-cells (as macrophages are non-dividing [67]), though we cannot rule out that these are *nef*-identical variants that infected large numbers of lung macrophages. Finally, some of our proviral dynamics interpretations depend on tree root placement, which is inherently uncertain [68, 69], though comparisons of on-ART proviral vs. pre-ART plasma HIV diversity are independent of root placement.

Our study nevertheless has some strengths, specifically the application of multiple compartmentalization tests, with each method employing a different level of complexity (SM being the most complex and AMOVA the least). This allowed us to classify datasets as compartmentalized (or not) with more confidence. A brief discussion of the advantages and limitations of each test is also useful. The tree-based methods SM [48] and CC [49] have the advantage of leveraging the evolutionary relationships among the sequences as informed by the tree, and generally have more power to detect compartmentalization [45]. These approaches however assume a high level of certainty in the underlying phylogeny, which is often not the case for within-host trees. To mitigate this, we integrated these tests over a Bayesian sampling [70] of 7,500 trees (indeed, the inherent uncertainty in phylogenetic inference is apparent in the consensus trees shown in S3, S4, S5, S6, and S7 Figs). It is also important to note that the tree-based tests consider topology (branching patterns) only, not branch lengths (long branches would thus convey the same information as shorter ones, despite describing a greater genetic distance), and that SM relies on a migration model whereas CC merely compares the correlation of the distance between tips. The genetic distance-based tests, AMOVA [46] and $K_{ST}$ [47], have the advantage of being able to provide results even when a phylogeny is difficult to resolve [37]. The availability of archived plasma from as long as 18 years prior to proviral sampling allowed us to interpret proviral diversity in context of HIV's within-host evolutionary history for two participants, highlighting differences in proviral longevity (*e.g.* participant 4's overall proviral pool contained numerous ancestral sequences, but viruses re-emerging in plasma from the reservoir dated to near ART initiation), and providing evidence of reservoir re-seeding during treatment interruption.

In summary, our results reveal that lung proviral diversity in individuals receiving long-term suppressive ART broadly reflects that in blood. "True" genetic compartmentalization (*i.e.* the presence of distinct genetic variants in blood and lung) is not the norm, and when present

it was only of modest magnitude. More frequently, genetic compartmentalization manifested as a numerical imbalance of *nef*-identical sequences in blood and lung, likely driven by differential clonal expansion across sites, but even this was not observed in everyone. More broadly, our study highlights the genetic complexity of HIV proviruses persisting in lung and blood during ART, and the uniqueness of each individual's proviral composition. Personalized HIV remission and cure strategies may be needed to overcome this.

## Materials and methods

### Ethics statement

This study was approved by the Providence Health Care/University of British Columbia and Simon Fraser University research ethics boards. All participants provided written informed consent.

### Participants and sampling

We studied paired buffy coat ("blood") and bronchoalveolar lavage (BAL) ("lung") specimens from nine participants with HIV, eight males and one female, who had undergone bronchoscopy at St. Paul's Hospital in Vancouver, Canada. Buffy coats and BAL were stored at -80°C until use. Participants were a median 60 (IQR, 51–62) years of age, and had maintained viremia suppression on ART for a median of 9 years (IQR, 4–15) at the time of sampling. Two participants (8 and 9) had been diagnosed with chronic obstructive pulmonary disease; the remainder had no documented overt respiratory symptoms or lung disease. Buffy coats had been separated from whole blood as described previously [71], and an estimated 5–10 million peripheral blood mononuclear cells (PBMC) were studied presently. BAL was performed as previously described [72]. The lavage was spun down into two cell pellets, each resuspended in 1mL Cytolyt solution (Cytyc, Marlborough, MA). One of these aliquots was studied presently. Total BAL cell counts and immune cell phenotyping were available for four participants (4, 5, 8, and 9): cell counts in one BAL aliquot ranged from 2.15–13 million, of which 92–99% were macrophages and <1–7% were lymphocytes. Longitudinal pre-ART plasma samples were also available from two participants (4 and 6).

### HIV amplification and sequencing

DNA was extracted from buffy coat and BAL using the QIAamp DNA Mini Kit (Qiagen). Single-genome amplification of a subgenomic HIV region (*nef*) was performed using primers designed to amplify major HIV subtypes [41, 51]. Briefly, genomic DNA extracts were endpoint diluted such that <30% of the resulting nested polymerase chain reactions (PCR), performed using an Expand High Fidelity PCR system (Roche), would yield an amplicon. First round PCR primers were Nef8683F_pan (forward; TAGCAGTAGCTGRGKGRACAGATAG) and Nef9536R_pan (reverse; TACAGGCAAAAAGCAGCTGCTTATATGYAG). Second round PCR primers were Nef8746F_pan (forward; TCCACATACCTASAAGAATMAGACARG) and Nef9474R_pan (reverse; CAGGCCACRCCTCCCTGGAAASKCCC). Negative controls were included in every run. For the participants with archived plasma available, nucleic acids were extracted using the BioMerieux NucliSENS EasyMag system (BioMerieux, Marcy-l'Étoile, France). Next, cDNA was generated using NxtScript reverse transcriptase (Roche) using the reverse primer Nef9536R_pan, after which the cDNA was endpoint diluted and single-genome-amplified as described above. Amplicons were sequenced on a 3730xl automated DNA sequencer using BigDye (v3.1) chemistry (Applied Biosystems). Chromatograms were base called using Sequencer (v5.0/v5.4.6) (GeneCodes).

## Sequence alignment and phylogenetic inference

Sequences exhibiting genetic defects (including large insertions/deletions), nucleotide mixtures, hypermutation (identified using Hypermut [73]) were excluded, as were sequences exhibiting evidence of within-host recombination (identified using RDP4 v4.1) [74]. Sequences were aligned in a codon-aware manner using MAFFT v7.471 [75]. Alignments were inspected and manually edited in AliView v1.26 [76]. A maximum-likelihood between-host phylogeny was inferred using IQ-TREE2 [77] following automated model selection using ModelFinder [78]. Branch support values were derived from 1,000 bootstraps.

Within-host phylogenies were inferred using Bayesian approaches as follows. First, each within-host nucleotide alignment was reduced to distinct sequences only. The best-fitting substitution model for each dataset was determined using jModelTest v2.1.10 (listed in S1 Table) [79]. Next, Markov chain Monte Carlo (MCMC) methods were used to build a random sample of phylogenies per participant, without enforcing any molecular clock. Two parallel runs with MCMC chains of five million generations, sampled every 1,000 generations, were performed in MrBayes, v3.2.7 [80] using the best-fitting substitution model and model-specific or default priors. Convergence was assessed by ensuring that the deviation of split frequencies was <0.03, that the effective sampling size of all parameters was ≥200, and through visual inspection of parameter traces in Tracer, v1.7.2 [81]. The first 25% of iterations were discarded as burn-in, yielding 7,500 phylogenies per participant.

Identical sequences were then grafted back on to these trees using the add.tips function in R package phangorn, v2.8.1 [82] to generate two sets of trees: one containing distinct sequences *per compartment* and another containing all within-host sequences collected. Phylogenies were plotted using the R (v4.1.2) package ggtree, version 3.21. The tree with the highest likelihood, midpoint rooted in FigTree (v1.4.4) (http://tree.bio.ed.ac.uk/software/figtree/) is displayed for each participant. Node support values are derived from Bayesian posterior probabilities from the consensus trees, which are shown in S3, S4, S5, S6, and S7 Figs.

For participants 4 and 6, for whom pre-ART plasma HIV RNA sequences were available, MCMC runs were additionally performed on alignments of proviral (blood-lung) and plasma HIV RNA sequences using the best-fitting substitution model for each dataset as listed in S1 Table. Trees were rooted using an outgroup. As the ideal outgroup is distantly (but not too distantly) related to the ingroup, we rooted each participant's phylogeny using a sequence from the most closely-related participant in our study (see Fig 1C)—namely, participant 4's phylogeny was rooted using the closest sequence from participant 1, whereas participant 6's phylogeny was rooted using the closest sequence from participant 7. Outgroup rooting was performed using the evolutionary placement algorithm in RAxML, implemented with a custom script in R [83], after which the branch leading to the outgroup was pruned off. Consensus trees were generated from outgroup rooted trees using the *consensus* and *consensus.edges* function using ape and phytools package in R.

## Topology testing of phylogenies

We inferred phylogenies in a maximum-likelihood framework using IQTree v1.6.1 [84] from participant 4 and participant 6's sequences using the best-fitting nucleotide substitution model (S1 Table). For each participant, one phylogeny was inferred under the constraint that plasma HIV RNA sequences from the earliest time point were separated from the plasma sequences sampled at later time points (though not necessarily separated from proviral sequences). This constrained phylogeny was compared with an unconstrained phylogeny through comparison of the tree likelihoods using IQTree's '-au' command, which performs five topology tests: the resampling of estimated log-likelihoods (RELL) test [52], the Kishino-Hasegawa (KH) test

[53], the Shimodaira-Hasegawa (SH) test [54], the approximately unbiased (AU) test [55], and the Expected Likelihood Weights (ELW) test [56].

### Genetic compartmentalization and statistical analyses

Compartmentalization was assessed using two genetic distance-based tests: Analysis of Molecular Variance (AMOVA) [46] and Hudson, Boos and Kaplan's nonparametric test for population structure ($K_{ST}$) [47], and two tree-based tests: the Slatkin-Maddison test (SM) [48] and the Correlation Coefficient (CC) test [49]. $K_{ST}$ was run in Hyphy [85] using the TN93 genetic distance matrix and the TN93 substitution model [86]. AMOVA was implemented in the R package pegas, v1.1 [87] using the K80 substitution model [88]. For these tests, statistical significance was assessed via 1,000 permutation tests. The tree-based tests were applied to all 7,500 phylogenies inferred per participant, and the output statistics were averaged across all trees. This was done using the R package slatkin.maddison, v0.1.0 for the SM test [89], and using a custom R script for the CC test. A within-host dataset was classified as compartmentalized when at least one test *per type* returned statistically significant results. Tests were performed on distinct sequences per compartment, as well as on all within-host sequences.

Blood and lung proviral diversity was assessed using two methods. For each phylogeny, we first computed the mean within-host patristic distance between all pairs of distinct sequences per compartment, and then calculated the mean over all 7,500 trees (i.e., *grand mean patristic distances*). Second, we summed the edge lengths of all distinct sequences per compartment in each tree, and then calculated the mean over all 7,500 trees (i.e., *mean phylogenetic diversity*) [90]. Both metrics were computed using custom R scripts. All other statistical analyses were performed in Prism, v9.0 (GraphPad Software). All compartmentalization tests of significance were one tailed (all others two tailed) with $p < 0.05$ denoting statistical significance.

### Data deposition

The nucleotide sequences reported in this paper are available in GenBank (accession numbers: proviral DNA: OM963156—OM964037, OP346862—OP346877; HIV RNA: OM964038—OM964339). The four custom R scripts mentioned in the methods can be found at: https://github.com/brj1/HIVCompartmentalization. A source file containing the data plotted in Figs 1A and 1B, 2B and 2D, 3B and 3D, 4B and 4D, 5B and 5C, 6C and 6D, 7C and 7D, S1B and S1D, and S2B is provided in S1 Data file.

### Supporting information

**S1 Fig. Within-host phylogenies and compartmentalization results for participants 3 and 5.** Legend as in Fig 2.
(TIF)

**S2 Fig. Within-host phylogeny and compartmentalization results for participant 7.** Legend as in Fig 2.
(TIF)

**S3 Fig. Within-host consensus phylogenies for participants 2 and 6.** Consensus phylogeny derived from Bayesian inference of 7,500 trees per participant, rooted at the midpoint. Filled circles denote blood sequences; open circles denote lung sequences. Asterisks identify nodes supported by posterior probabilities $\geq 70\%$.
(TIF)

**S4 Fig. Within-host consensus phylogenies for participants 4 and 8.** Legend as in S3 Fig.
(TIF)

**S5 Fig. Within-host consensus phylogenies for participants 1 and 9.** Legend as in S3 Fig.
(TIF)

**S6 Fig. Within-host consensus phylogenies for participants 3 and 5.** Legend as in S3 Fig.
(TIF)

**S7 Fig. Within-host consensus phylogeny for participant 7.** Legend as in S3 Fig.
(TIF)

**S8 Fig. Within-host consensus phylogenies for participants 4 and 6.** Consensus phylogenies derived from Bayesian inference of 7,500 outgroup rooted trees per participant. Filled circles denote blood sequences; open circles denote lung sequences; diamonds show plasma sequences. Asterisks identify nodes supported by posterior probabilities $\geq 70\%$.
(TIF)

**S9 Fig. Summary of genetic compartmentalization results from two batches of sequences within the same compartment within a participant.** Summary of genetic compartmentalization results for the five participants for whom proviral sequences were collected from two independent aliquots of the same material, where a dataset is declared compartmentalized if at least one test per type (genetic distance vs. tree-based) gave a statistically significant result (see "consensus" column). The number of sequences recovered from each blood or lung aliquot is indicated in the participant label. Dark grey squares denote compartmentalization with $p<0.001$ (no test reached this threshold), light grey squares denote compartmentalization with $p<0.05$; white squares denote no compartmentalization. The "Distinct" panel summarizes compartmentalization results when limiting analysis to distinct sequences per compartment; the "Overall" panel summarizes results when all sequences are included.
(TIF)

**S1 Table. Best-fit models of nucleotide substitution for each within-host dataset.**
(DOCX)

**S1 Data. Source file for data plotted in graphs.** Data values used to generate the graphs shown in Figs 1A and 1B, 2B and 2D, 3B and 3D, 4B and 4D, 5B and 5C, 6C and 6D, 7C and 7D, S1B and S1D, and S2B (each graph's source data is on a separate tab).
(XLSX)

## Acknowledgments

We sincerely thank the study participants without whom this research would not be possible.

We thank Mark Brockman for helpful discussions. We thank Hanwei Sudderuddin, Hope Lapointe and Sarah Speckmaier for technical assistance.

## Author Contributions

**Conceptualization:** Aniqa Shahid, Bradley R. Jones, Jeffrey B. Joy, Janice M. Leung, Zabrina L. Brumme.

**Data curation:** Aniqa Shahid, Julia S. W. Yang, Kathryn Donohoe, Chanson J. Brumme.

**Formal analysis:** Aniqa Shahid, Bradley R. Jones.

**Funding acquisition:** Jeffrey B. Joy, Zabrina L. Brumme.

**Investigation:** Aniqa Shahid, Winnie Dong.

**Methodology:** Aniqa Shahid, Bradley R. Jones, Jeffrey B. Joy.

**Project administration:** Aniqa Shahid, Zabrina L. Brumme.

**Resources:** Tawimas Shaipanich, Chanson J. Brumme, Jeffrey B. Joy, Janice M. Leung, Zabrina L. Brumme.

**Software:** Bradley R. Jones.

**Supervision:** Jeffrey B. Joy, Janice M. Leung, Zabrina L. Brumme.

**Validation:** Aniqa Shahid, Zabrina L. Brumme.

**Visualization:** Aniqa Shahid.

**Writing – original draft:** Aniqa Shahid, Zabrina L. Brumme.

**Writing – review & editing:** Aniqa Shahid, Bradley R. Jones, Julia S. W. Yang, Winnie Dong, Tawimas Shaipanich, Kathryn Donohoe, Chanson J. Brumme, Jeffrey B. Joy, Janice M. Leung, Zabrina L. Brumme.

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
