## [Decision Letter · Decision Letter 0]

12 Jul 2022

Dear Dr. Brumme,

Thank you very much for submitting your manuscript "HIV proviral genetic diversity, compartmentalization and inferred dynamics in lung and blood during long-term suppressive antiretroviral therapy" for consideration at PLOS Pathogens. As with all papers reviewed by the journal, your manuscript was reviewed by members of the editorial board and by several independent reviewers. In light of the reviews (below this email), we would like to invite the resubmission of a significantly-revised version that takes into account the reviewers' comments.

There are several suggestions we offer in addition to the comments of the reviewers.

1. The 3 samples with low sequence numbers from the lung are a distraction (as the reviewers noted). When presented as equivalent data to the more heavily sampled people this weakens the argument you make with the other samples. An alternative approach would be to not include them or include them in a more cursory way as confirmatory with data not shown (or move to supplemental material).  The optimal alternative would be to generate more sequences from those three subjects (3, 5, 7). With so few sequences from these three people about all you can say is if the viral sequence distribution is either highly clustered or diverse with the rest of the blood sequences. For #7 it might be possible to do a resampling simulation to determine if the absence of detection of the large clone in the blood is a statistically significant unlikely outcome given the sample size in the lung (which would show asymmetric distribution of this large blood clone for not being in the lung).

2. Is there a reason some of the samples have so few sequences? If one wanted to repeat your work then it would be helpful to know how many cells were in each BAL sample (did you get/use all of the sample?). Alternatively how much DNA, and what was the HIV DNA to cell DNA ratio. One could imagine a scenario where clonal amplification is driving the HIV copy number so samples with little clonal amplification give diverse populations but with low DNA copy numbers. Is this what is going on?

Beyond these comments the reviewers make a number of relevant points that need to be carefully considered.

We cannot make any decision about publication until we have seen the revised manuscript and your response to the reviewers' comments. Your revised manuscript is also likely to be sent to reviewers for further evaluation.

Sincerely,

Ronald Swanstrom

Associate Editor

PLOS Pathogens

Alexandra Trkola

Section Editor

PLOS Pathogens

Kasturi Haldar

Editor-in-Chief

PLOS Pathogens

orcid.org/0000-0001-5065-158X

Michael Malim

Editor-in-Chief

PLOS Pathogens

orcid.org/0000-0002-7699-2064

Reviewer's Responses to Questions

**Part I - Summary**

Reviewer #1: Shahid et al examine viral diversity in the lung and blood during ART. This is an extremely interesting and understudied topic. Based on DNA-derived nef sequences from the blood and lungs during suppressive ART and RNA-derived sequences from the blood during viremia, the authors make a number of interesting observations including the following. First, they do not observe extensive compartmentalization in the lungs, suggesting that prior to ART, viral populations do not replicate extensively at this site independent of variants in the blood. Second, they observe identical nef sequences in the blood and lungs further illustrating mixing between these compartments. Finally, they observe a higher percentage of clonal sequences in the lungs than blood. Together these results are consistent with viral dynamics in the lungs possibly being driven by inflammatory processes generating migration of infected cells from the periphery to the lungs and clonal expansion in the lungs.

Unfortunately the study has a number of issues that limit the strength of these (and other) conclusions. One unsurprising issue is that the authors typically generated few sequences from the lungs and much larger numbers from the blood. As a result, it is difficult to define features of viral population in the lungs and assess compartmentalization. Second, some of their phylogenetic approaches may not be appropriate for the complex samples that they are analyzing. For participant 6 they constructed maximum likelihood trees whose root was intended to be the MRCA. However, because of the participants unusual suppression history and a lack of longitudinal pre-ART samples, their approach rooted the tree to sequences from an intermediate timepoint despite the tree containing RNA-derived sequences from a much earlier timepoint. This violates the assumption that the root is the MRCA of the population. These concerns make it challenging to generate conclusions about viral dynamics in the lungs.

Reviewer #2: Shahid et al. sequenced the nef genomic region of proviruses within cells from the lung and compared these nef sequences from the lung to proviral sequences from blood-derived cells in nine participants on long-term therapy. They found identical sequence expansions varied across participants and that 77% of the participants had identical proviral sequences from the lung and blood anatomic sites. The authors employed four methods for measuring compartmentalization of proviral sequences between the lung and blood anatomic sites and observed little evidence for lung-blood compartmentalization of proviral sequences in 5 out of the 9 participants. However, they did find large differences in the proviral dynamics as exhibited by the proportion of persisting proviruses that represented ancestral versus more recently-circulating viral sequences. This is a well-written and interesting study focusing on an understudied anatomic site, the lung, and provides important genetic analysis of the proviruses found in this site compared to the blood, However the are some limitations of the study which the authors should address and discuss.

Reviewer #3: Shahid and colleagues compare HIV proviral populations in blood and lung in individuals with viremia suppressed on long-term ART. The authors describe how their study differs from previous work and adds to our understanding of HIV persistence. They demonstrate modest compartmentalization in the lung of some people living with HIV that likely results from different frequencies of infected T cells clones and from infection of lung macrophages. The study supports the idea that, overall, measuring the HIV reservoir in blood is a good representation of virus that may be found in tissues such as lung (presented here).

Suggestions:

1) In the Author Summary, change last sentence from "Critically, the extent of within-host proviral diversity in blood correlated strongly with that in lung, indicating that despite inter-individual heterogeneity, blood is a strong indicator of proviral diversity elsewhere in the body" to "Critically, the extent of within-host proviral diversity in blood correlated strongly with that in lung, indicating that despite inter-individual heterogeneity, blood is a strong indicator of proviral diversity in the lung" since other tissues were not evaluated. Some studies have shown good evidence of compartmentalization in brain.

2) Table 1- 1) Change word "intact" to "nef". Intact infers full-length HIV sequencing. 2) Add estimated duration of infection prior to ART initiation

3) Remove the word "intact" through the text since it implies full-length HIV sequencing. Use “nef” or sub-genomic” instead throughout.

4) Please review the statistics for the compartmentalization tests. It does not seem that they were corrected for multiple comparisons. For example, it is typical for the KST to be significant when the p value is <10-3 when applied to single-genome sequencing data to correct for sampling error of highly diverse HIV populations in vivo. I am not sure if the p value of <0.05 was used appropriately for these tests.

5) The statement in line 365 that claims that the data shows that clonal populations of infected cells are not present in all patients should be removed. When identical sequences are not found, it does not mean that clones of infected cells are not present since each of the sequences detected could be part of small cell clones that were detected only once at the level of sampling in the study.

6) The word "clades" should be changed to "variants" in line 390.

7) I suggest adding more discussion on how sampling affects the analyses. For example, if you had sampled sequences in the blood or lungs "twice", could occasional "compartmentalization" be seen in different donors just because few proviral sequences are obtained relative to all the infected cells in the body? I would also further discuss the effect of shallow sampling of the lungs, emphasizing that the large rakes of identical sequences in the lungs are likely very large infected cell clones or identical variants that infect lung macrophages.

8) I would remove the lines connecting the points in Figure 1B and, instead, add a bar at the median values since, I think, that is what the p value is referring.

9) Line 150: “…highest likelihood phylogeny from among 75,000 within-host reconstructions is shown.” Some justification should to be added for why the authors select the highest likelihood topology. Typically, the consensus tree from the Bayesian analysis is used. Selecting the highest likelihood could introduce bias as it doesn’t represent the Maximum clade credibility (>95%) of the 75,000 within-host reconstructions. Same comment for Line 519.

10) Lines 382-384: “Overall, our results indicate that compartmentalization is generally limited, of modest magnitude, and when present it is often due to differential distributions of clonally-expanded populations.” A discussion of the pros/cons for the statistical tests used might be useful.

11) Line 435: A parenthesis needs to be added.

12) Line 502: Did ModelFinder demonstrate any uniquely different models for each PID, or was it consistent? If so, might be worth including.

13) Line 506: Where ModelFinder and jModelTest in agreement?

14) Line 509-510: Without the statement, is it to be assumed that no clock was enforced? But if it was, the arithmetic/harmonic means of the null and +clock should be included as well as an explanation for its use. Since the Bayesian analysis was not conducted in BEAST, it appears that no clock was enforced.

15) Line 542-547: How does the patristic distances differ from an APD calculation? The authors report the percent of distinct sequences and genetic distance within each compartment, but the explanation of diversity between

**Part II – Major Issues: Key Experiments Required for Acceptance**

Reviewer #1: 1) There are a number of issues with the compartmentalization analyses that should be considered.

• Uneven and small sample sizes greatly reduce the power and accuracy of compartmentalization analyses (Zarate et al 2007). In this study 4 of 9 participants had fewer than 10 sequences from the lungs and an average of 75 sequences from the blood. The small number of lung sequences and the very large number of blood sequences makes it impossible to accurately assess compartmentalization for these individuals. This problem is worse when the analysis is restricted to distinct sequences in which case 6 of 9 participants had fewer than 10 sequences from the lungs and an average of 48 sequences from the blood. The primary finding of the paper is that there isn't evidence of major compartmentalization in the lungs, however, the limited sampling reduces the power of the analyses.

• Given Zarate's observation that distance and tree-based methods have different strengths and weaknesses, a more conservative approach would be to declare compartmentalization significant if at least one test of each type is significant.

• The rationale behind the "distinct" analysis, i.e. "where any sequence present in both lung and blood was represented once per compartment" is unclear to me. This suggests that if multiple identical sequences were observed in one compartment all of those sequences would be included in the analysis, but if they were present in two compartment only a single sequence would be included from each. Is that correct? By analyzing multiple identical sequences within but not across the compartment, wouldn't that bias the analyses toward identifying compartmentalized lineages?

2) I understand the general rooting approach that the authors used for their trees containing plasma RNA, however, I'm not sure how intermittent suppression impacts this approach. The authors indicate that in regard to participant 6, "We therefore rooted each of the participant's 7,500 phylogenies at the location that maximized the correlation between root-to-tip distances and sampling time of plasma sequences collected in 2006 and 2009." However, the tree also contains pre-ART sequences from 1996. In addition, the reservoir is likely to be very diverse and all of the reservoir may not reactivate at the same time. Could this artificially increase the amount of diversity that accumulated between 2006 and 2009 and over-estimate the rate of evolution? It would be nice to include a more thorough discussion of this analysis and the impact that intermittent suppression may have on it.

3) "Assuming that the root represents the MRCA of the sequences collected, we can make some inferences about proviral dynamics." I don't see how this can be true. In figure 8 the root to tip distance was higher for samples collected in 1996 than samples collected in 2006. Thus the root can't be the MRCA of all the sequences in the tree. This approach is inappropriate for this dataset and causes the tree to misrepresent the within host evolutionary dynamics. The authors state " This strategy works because evolving within-host HIV populations exhibit increasing divergence from the root over time." However, the oldest sequences in the tree are not at the root. Given these concerns it is unclear that any conclusions can be made from the trees in figures 8 and 9. The authors should consider an alternative approach to analyses of sequences from these participants.

4) The authors should clarify the goals of examining compartmentalization in this cohort. With a few exceptions, studies have not found evidence of ongoing viral replication during ART, thus compartmentalization analyses in people on ART are typically examining whether distinct populations were established in a compartment prior to ART. The authors indicate that participant 9 initiated ART soon after diagnosis thus limiting their ability to assess viral replication in the lungs pre-ART. Another interesting question is whether HIV-infected cells migrate between the blood and tissues during ART. However, addressing this hypothesis may require knowledge of the population in the lungs before ART. The authors should specify the hypotheses that they are testing and the type of data that can be used to address those hypotheses.

5) Given the small number of lung sequences obtained for many of the participants, it is difficult to accurately estimate parameters about those sequences (percent distinct, hypermutated, intact, etc). As a result, it's difficult to make generalizations about differences across participants (See Fig 1b) and compartments (See 7b and 7c). This should be mentioned whenever discussing differences.

6) Is it appropriate to compare divergence from the root in participants 4 and 6 despite the fact that one was rooted using HXB2 and the other using RNA derived sequences from that participant?

7) For clarity, please specify that you are looking at nef sequences in table 1 (e.g. "intact nef DNA sequences from the lung ")

Reviewer #2: 1) The authors state several limitations of their study in their discussion section of the manuscript but they do not address whether the limited number of proviral sequences from the lung compartment for 4 participants (<10 sequences in total) affected the accuracy of the methods used to determine compartmentalization between the lung and blood anatomic sites. For many compartmentalization methods if one site has approximately 10-fold more sequences (ie the blood-derived proviral sequences), this can affect the accuracy of the methods used to calculate compartmentalization. The authors should discuss in their results section whether the disparity in the number of proviral sequences between the lung and blood compartments could affect their results. If so, how did they address this issue?

2) An elegant study of the proviral diversity in context of pre-ART viral populations was presented for two participants. However, in this section the comparison of the pre-ART and treatment interruption plasma-derived sequences to the proviral sequences in the lung is not described in-depth. For each participant, the authors should add a small section as to how the plasma-derived RNA sequences compared to the lung proviral sequences. Is there any evidence or indication as to when the lung anatomic site was seeded? Was this site seeded prior to therapy initiation or during the treatment interruption phase?

3) The authors found that the diversity of distinct proviral sequences recovered from blood correlated significantly with the diversity of distinct sequences recovered from lung for these participants. Although this was calculated using two methods, if the number of distinct sequences falls below 5, this can affect the accuracy of diversity calculations. Did any of the participants have less than 5 distinct sequences from the lung? If so, the authors should graph their data with and without these participants to determine if participants with low numbers of distinct proviral sequences from the lung affects the correlation of distinct proviral sequences between the lung and blood compartments.

4) In this study the authors sequence the nef region of pre-therapy and treatment interruption plasma-derived RNA. A recent full-length HIV RNA study has revealed that plasma-derived sequences are not all intact and that up to 50% are defective. The authors should describe how sequencing the nef region of HIV RNA may be a limitation and how this could affect their findings especially if this does not always represent a full-length RNA sequence or replication-competent virus.

Reviewer #3: (No Response)

**Part III – Minor Issues: Editorial and Data Presentation Modifications**

Reviewer #1: 1) The duration of untreated infection should be designated in Table 1.

2) It's interesting that some participants primarily had clonal viruses in their lungs at the on-ART timepoint. Is there any evidence that they had a respiratory infection or other illness that could drive clonal expansion?

Reviewer #2: This is only semantics but when referring to intact proviral sequences please change this to nef-intact proviral sequences.

Reviewer #3: (No Response)

PLOS authors have the option to publish the peer review history of their article (what does this mean?). If published, this will include your full peer review and any attached files.

Reviewer #1: No

Reviewer #2: No

Reviewer #3: No
---

## [Decision Letter · Decision Letter 1]

21 Oct 2022

Dear Dr. Brumme,

We are pleased to inform you that your manuscript 'HIV proviral genetic diversity, compartmentalization and inferred dynamics in lung and blood during long-term suppressive antiretroviral therapy' has been provisionally accepted for publication in PLOS Pathogens.

Best regards,

Ronald Swanstrom

Associate Editor

PLOS Pathogens

Alexandra Trkola

Section Editor

PLOS Pathogens

Kasturi Haldar

Editor-in-Chief

PLOS Pathogens

orcid.org/0000-0001-5065-158X

Michael Malim

Editor-in-Chief

PLOS Pathogens

orcid.org/0000-0002-7699-2064

Reviewer Comments (if any, and for reference):

Reviewer's Responses to Questions

**Part I - Summary**

Reviewer #1: (No Response)

**Part II – Major Issues: Key Experiments Required for Acceptance**

Reviewer #1: This manuscript went through a very rigorous review process and the authors made considerable effort to address the reviewers' concerns. The revised product is unique and fairly represents their findings and the limitations of those findings. I am particularly appreciative of them performing their analyses with and without the individuals with poor sampling and the thoroughness of their statistical and phylogenetic analyses.

**Part III – Minor Issues: Editorial and Data Presentation Modifications**

Reviewer #1: (No Response)

PLOS authors have the option to publish the peer review history of their article (what does this mean?). If published, this will include your full peer review and any attached files.

Reviewer #1: No

---

## [Editor Report · Acceptance letter]

31 Oct 2022

Dear Dr. Brumme,

We are delighted to inform you that your manuscript, "HIV proviral genetic diversity, compartmentalization and inferred dynamics in lung and blood during long-term suppressive antiretroviral therapy," has been formally accepted for publication in PLOS Pathogens.

Best regards,

Kasturi Haldar

Editor-in-Chief

PLOS Pathogens

orcid.org/0000-0001-5065-158X

Michael Malim

Editor-in-Chief

PLOS Pathogens

orcid.org/0000-0002-7699-2064